# Miniaturized Low-Frequency Communication System Based on the Magnetoelectric Effect

**DOI:** 10.3390/mi14101830

**Published:** 2023-09-26

**Authors:** Guohao Zi, Zhibo Ma, Yinan Wang, Yuanhang Wang, Ziqiang Jia, Shanlin Zhao, Dishu Huang, Tao Wang

**Affiliations:** 1The Ministry of Education Key Lab of Micro/Nano Systems for Aerospace, Northwestern Polytechnical University, Ministry of Education, Xi’an 710072, China; zigh@mail.nwpu.edu.cn (G.Z.); wangyinan@mail.nwpu.edu.cn (Y.W.); wwyyhh@mail.nwpu.edu.cn (Y.W.); jiaziqiang@mail.nwpu.edu.cn (Z.J.); zhaoshanlin@mail.nwpu.edu.cn (S.Z.); huangdishu@mail.nwpu.edu.cn (D.H.); 2Shaan’xi Key Lab of MEMS/NEMS, Northwestern Polytechnical University, Xi’an 710072, China; 3Ningbo Research Institute, Northwestern Polytechnic University, Ningbo 315100, China

**Keywords:** low-frequency communication system, miniaturization, magnetoelectric antenna, underwater communication

## Abstract

Recently, the realization of electromagnetic wave signal transmission and reception has been achieved through the utilization of the magnetoelectric effect, enabling the development of compact and portable low-frequency communication systems. In this paper, we present a miniaturized low-frequency communication system including a transmitter device and a receiver device, which operates at a frequency of 44.75 kHz, and the bandwidth is 1.1 kHz. The transmitter device employs a Terfenol-D (80 mm × 10 mm × 0.2 mm)/PZT (30 mm × 10 mm × 0.2 mm)/Terfenol-D glued composite heterojunction magnetoelectric antenna and the strongest radiation in the length direction, while the receiver device utilizes a manually crafted coil maximum size of 82 mm, yielding a minimum induced electromagnetic field of 1 pT at 44.75 kHz. With an input voltage of 150 V, the system effectively communicates over a distance of 16 m in air and achieves reception of electromagnetic wave signals within 1 m in simulated seawater with a salinity level of 35% at 25 °C. The miniaturized low-frequency communication system possesses wireless transmission capabilities, a compact size, and a rapid response, rendering it suitable for applications in mining communication, underwater communication, underwater wireless energy transmission, and underwater wireless sensor networks.

## 1. Introduction

Wireless communication technologies have significantly enhanced industrial production efficiency and greatly improved the quality of daily life. In modern society, radio frequency (RF) technology-based wireless communication, particularly Bluetooth [1,2] communication in the ISM (Industrial, Scientific, Medical) band and millimeter wave communication exemplified by WiFi [3] and 5G [4,5] plays a crucial role. High-frequency electromagnetic waves carry dense information, but they are prone to interference and quickly attenuate in highly conductive media. Conversely, low-frequency electromagnetic waves (30–300 kHz) exhibit strong resistance to interference, possess a large skin depth, and demonstrate excellent penetration capabilities in highly conductive media, particularly in underwater communication and ground-penetrating communication, thus showcasing promising applications [6,7]. However, low-frequency electrical antennas are hindered by the matching relation between the physical size and wavelength, resulting in their substantial dimensions, and shrinking the antenna size reduces its gain [8,9], significantly limiting the portability of low-frequency communication systems. Consequently, urgent research is needed to develop miniaturized, lightweight, and low-power low-frequency communication systems through innovative approaches. In the ISM band, the metamaterial [10,11] and metasurfaces [12,13] are important methods to improve the performance parameters of antenna devices and to design an antenna in a compact footprint area. However, a key method to reduce the size of low-frequency antennas is replacing electrical antennas with mechanical antennas [14]. Unlike conventional electrical antennas that rely on electrons generating oscillating currents in circuits to radiate electromagnetic waves, mechanical antennas employ mechanical energy to drive the motion of electric charges or magnetic dipoles, thus generating a radiation field. This approach enables the utilization of near-field energy, which is challenging to harness in conventional electrical antennas, in antenna radiation and facilitates the miniaturization of low-frequency communication devices without requiring extensive impedance matching networks [15]. Mechanical antennas can be categorized into four types: electret type [16,17], permanent magnet type [18], piezoelectric type [19], and magnetoelectric type [20,21,22]. Among these, magnetoelectric mechanical antennas have gained popularity in recent years due to their low power consumption, light weight, and rapid response.

Magnetoelectric heterojunctions have found applications in tunable inductors [23], tunable filters [24], magnetoelectric memories [25], energy harvesting devices [26], and magnetic sensors [27]. In 2017, Nan et al. designed an acoustically actuated nanomechanical magnetoelectric (ME) antenna operating at a frequency of 60.7 MHz, based on the magnetoelectric effect [28], and experimentally verified the feasibility of using magnetoelectric heterojunctions in antennas. This study demonstrated that the magnetoelectric structure enables the transformation of electromagnetic fields and oscillating currents without being limited by the resonance of the antenna size and specific electromagnetic wave wavelengths. Consequently, the limitation imposed by the electromagnetic wave wavelength on antenna size can be overcome. Dong et al. reduced the operating frequency of the magnetoelectric (ME) antenna to 23.95 kHz [29]. They successfully achieved signal transmission and reception using ME antennas and found that the radiation pattern of the ME antenna conformed to the standard magnetic dipole radiation equation. The ME antenna operates on the principle that, as a transmitting antenna, an alternating power source stimulates the piezoelectric layer to generate vibrations. These vibrations are then transmitted through the heterojunction to the magnetostrictive layer, causing magnetization oscillation that radiate electromagnetic waves. As a receiving antenna, the magnetostrictive layer senses spatial electromagnetic waves and strains, which are conveyed by the heterojunction to the piezoelectric layer. The piezoelectric layer outputs a voltage signal as a result of the inverse piezoelectric effect.

Xu et al. conducted simulations and confirmed that the near-field coupling between the ME antenna and the coil depends on mutual inductance rather than electromagnetic radiation [30]. Hu et al. discovered that the resonant frequency of the ME antenna can be adjusted by modifying the constraint [31]. Furthermore, for multilayer thin sheet composite heterojunctions, optimizing the interlayer bonding can effectively enhance the magnetoelectric coupling coefficient [32], and the application of a DC bias magnetic field can improve the energy conversion efficiency between magnetostrictive and piezoelectric materials [33]. Taking it a step further, Niu et al. utilized small permanent magnets instead of Helmholtz coils to provide a DC bias magnetic field for ME antennas, resulting in improved energy conversion efficiency while maintaining a compact size [34,35].

This paper presents the design of a low-frequency miniaturized magnetoelectric communication system (i.e., MMCS) including transmitter device and receiver device, where the transmitting device is a ME antenna and the receiving device is a manually crafted coil. In this paper, we first describe the design and manufacture of transmitter device and receiver device of MMCS and then the building of a low-frequency electromagnetic wave test platform to complete the parameter characterization of MMCS and analysis test results. The transmitting ME antenna was constructed as a Terfenol-D/PZT/Terfenol-D three-layer magnetoelectric structure with dimensions of 80 mm × 10 mm × 3 mm. To enhance the transmitting signal strength, three layers of 30 mm × 10 mm × 2 mm NdFeB magnets were integrated to improve the stress transfer efficiency between the heterojunction layers and the strongest radiation in the length direction. The receiving coil was wound with 0.3 mm copper enameled wire, featuring an inner diameter of 46 mm, an outer diameter of 82 mm, an axial length of 40 mm, and a minimum resolution of 1 pT. The MMCS operated at a frequency of 44.75 kHz that carried on the binary code element transmission, and the experimental results indicated an effective communication distance of 16 m in an air medium. Compared with similar research literature [36,37], this device was able to detect smaller magnetic fields due to the resonant frequency at which the receiver coil operates, which helps to improve the actual test communication distance. In a simulated 35% salinity seawater medium, the receiving coil located 1 m away from the transmitting antenna successfully captured discernible electromagnetic waves. The MMCS exhibits a wide range of potential applications in underwater communication, underwater energy harvesting, and mining communication.

## 2. MMCS Design and Manufacturing

Figure 1a shows the schematic diagram of the transmitting ME antenna, operating in the L-T vibration mode [38], scilicet, the piezoelectric material, is polarized longitudinally, and the magnetostrictive material is magnetized transversely. The AC power supply excites the transmitting ME antenna, applying the excitation to the piezoelectric layer (i.e., PZT), whose components are lead–zirconate–titanate piezoelectric ceramics. The inverse piezoelectric effect in the PZT generates periodic vibrations, which are transmitted to the magnetostrictive material layer (i.e., Terfenol-D). The periodic vibration of Terfenol-D produces an electromagnetic wave signal at the excitation frequency, enabling the radiation of electromagnetic waves.

The ME antenna has a sandwich structure of Terfenol-D/PZT/Terfenol-D. The Terfenol-D layer (80 mm × 10 mm × 0.2 mm) was sourced from Shijiazhuang Saining Electronic Technology Co. (Shijiazhuang, China), and the PZT layer (30 mm × 10 mm × 0.2 mm) was sourced from the Zhejiang Shenlei Ultrasonic Material Factory. The fabrication process of the transmitting ME antenna involved several steps. First, a c-axis polarized PZT piezoelectric sheet measuring 30 mm × 10 mm × 0.2 mm was cut, along with two Terfenol-D magnetostrictive sheets measuring 80 mm × 10 mm × 0.2 mm. Next, a pair of PI (i.e., Polyimide) substrate electrodes with a total width of 10mm and a length of 30 mm, printed with purple copper electrodes of 20 um thickness, were prepared. The PI substrate was positioned between the piezoelectric and magnetostrictive sheets, with the printed electrode surface facing the piezoelectric material. Devcon14250 epoxy resin adhesive was used for interlayer bonding, and the antenna was subjected to 0.1 MPa pressure on a bonding machine for proper adhesion. Afterward, the antenna was left to cure for 24 h. The positive and negative inputs were connected to the upper and lower copper electrodes, respectively, to interface with the excitation source. The final ME antenna, shown in Figure 1b, measured 80 mm × 10 mm × 3 mm. To enhance its radiation performance, three layers of small NdFeB magnets measuring 30 mm × 10 mm × 2 mm were integrated to improve the energy transfer efficiency.

The receiving device consists of a manually crafted coil wound with 0.3 mm copper enameled wire, comprising 800 turns. The coil has an inner diameter of 46 mm, an outer diameter of 58 mm, and an effective axial length of 12 mm. Figure 1c illustrates the 3D-printed carbon fiber reinforced polyester frame of the receiving coil, measuring Φ82 mm × 40 mm in total volume. The output voltage is led out through a bayonet nut connector (BNC) connector.

## 3. Experimental Platform

The principle of the MMCS receiving coil characterization platform is illustrated in Figure 2. To minimize the impact of environmental noise, the calibration experiment for the receiving coil was conducted inside an electromagnetic field shielding cylinder, which remained closed throughout the test. A precision current source (Keithley-6221) was used to supply input current to the Helmholtz coil, generating the electromagnetic field. The receiving coil was connected to a lock-in amplifier (Stanford Research Systems SR830), which measures the induced voltage output from the receiving coil. The electromagnetic field intensity at the center of the Helmholtz coil was measured using a low-frequency electromagnetic signal analyzer (NF-5035) manufactured by Aaronia AG, Euscheid, Germany. By plotting the relationship between the spatial electromagnetic field and the receiving coil’s output voltage, we analyzed the performance of the receiving coil.

The test setup for the MMCS is depicted in Figure 3. A waveform generator (Tektronix AFG3022C) generates a sine wave signal that serves as the excitation source. The excitation signal was amplified by a power amplifier (Aigtek ATA-3080). The amplified signal was then applied to the transmitting ME antenna, which emits electromagnetic waves at the same frequency as the excitation signal. The receiving coil captured the electromagnetic waves, inducing an output voltage signal. This voltage signal was either fed into the lock-in amplifier or the oscilloscope (Rigol-DS2102) for signal analysis.

## 4. Results

### 4.1. Receiving Coil Characterization

The characterization platform for the MMCS receiving coil is illustrated in Figure 4a. The resonant frequency of the MMCS receiving coil was 44.75 kHz, as depicted in Figure 4b. At this resonant frequency, the receiving coil exhibited the strongest signal, leading to improved detection resolution and enhanced communication distance for the MMCS. And the bandwidth of the receiving coil was 1.1 kHz; it needs to be clarified that the difference between the two frequencies corresponding to the value of the resonant peak falling $\sqrt{2}/2$ times its maximum value is defined as the bandwidth.

Figure 4 shows the receiving coil operating frequency at 44.75 kHz; within the effective measurement range, an electromagnetic field strength of 1 nT corresponds to an output voltage of 86.446 mV. The receiving coil was capable of detecting AC electromagnetic fields as low as 1 pT. Signals below 1 pT were submerged in the noise, as shown in Figure 4d.

### 4.2. Transmitting ME Antenna Experiment

Figure 5 illustrates the relationship between the radiated electromagnetic wave intensity of the transmitting ME antenna and the peak of the input voltage. It is worth emphasizing that the voltages mentioned in this paper represent only the amplitude of the sine wave. The transmitting ME antenna requires a starting voltage, and when the excitation voltage is below 3 V, the vibration of the PZT sheet is weak, resulting in a weaker radiated signal compared to ambient noise. As the input voltage exceeds 3 V, the signal from the transmitting ME antenna grows significantly with the voltage. However, this growth is not limitless. Beyond a voltage of 200 V, the vibration of the piezoelectric sheet reaches saturation, and further increasing the excitation voltage does not produce a stronger output electromagnetic wave signal. It is important to note that the starting and saturation voltages may vary depending on the size, shape, structure, and material components of the piezoelectric layer [31,35].

Figure 6 presents the impact of the integrated small NdFeB magnets on the radiation signal of the transmitting ME antenna. Each individual magnet has a volume of 30 mm × 10 mm × 2 mm. The test results demonstrated that the integration of small NdFeB magnets helped to enhance the antenna’s radiation strength. This enhancement is attributed to the pressure effect and DC bias magnetic field because the application of pressure improves the stress transfer between the heterojunction layers, and Terfenol-D is effectively polarized by the DC bias magnetic fields, both leading to more efficient energy conversion. Additionally, it is important to mention that although the near-field radiation intensity of the ME antenna was significantly enhanced as the number of magnets increased, this radiation intensity decayed rapidly with increasing distance. This is because the attenuation of electromagnetic waves in the near field is inversely proportional to the cube of the distance [39].

The radiation pattern of the ME is helpful to grasp the magnetic field intensity distribution around the ME antenna and provide guidance for the placement of the ME antenna. Firstly, we define the coordinate system of the ME antenna, as shown in Figure 7a, the length direction, width direction, and thickness direction of ME antenna are defined as the *x*-axis, *y*-axis, and *z*-axis, respectively. The *x*-axis direction is defined as 0°, and the *y*-axis direction is defined as 90°. Since the experiment described in this paper was carried out on the *x*-*y* plane, it mainly characterizes the radiation pattern of the *x*-*y* plane of the ME antenna. In order to reduce the influence of the power supply line on the test results, the receiving device was 0.1 m away from the ME antenna. The experimental results of the normalization processing are shown in Figure 7b. The radiation intensity was the highest in the directions of 0° and 180°, and the radiation intensity was the lowest in the directions of 90° and 270°. That is to say, in the *x*-*y* plane, the radiation in the length direction was the strongest, and the radiation in the width direction was the weakest. In the experiments described below, it was necessary to ensure that the length direction of the ME antenna was aligned with the receiving device.

### 4.3. MMCS Communication Performance Test

When the waveform generator generated a standard sine wave at 44.75 kHz, which is the resonant frequency of the MMCS, the amplified excitation signal was applied to the transmitting ME antenna. As a result, the antenna radiated an electromagnetic wave into the surrounding space. Simultaneously, the receiving coil captured the electromagnetic wave and produced an output voltage that remained a sine wave at 44.75 kHz, as depicted in Figure 8. This output signal from the receiving coil was consistent with the radiation of the transmitting ME antenna.

As a communication system, it is necessary for the MMCS to carry out binary code element transmission experiments. The distance between the transmitting device and the receiving device was 3 m, and the AM modulation signal was transmitted. The carrier frequency was 44.75 kHz, the modulation signal was a 200 Hz sine wave, and the modulation depth was 100%. The modulation waveform and the resolution of the modulation information are shown in Figure 9, which indicates that MMCS has the potential for communication.

The MMCS operates in the low-frequency band within the near-field region (r << λ), where r represents the distance between the transmitting ME antenna and the receiving coil, and λ denotes the wavelength of the electromagnetic wave. As shown in Figure 10, an experimental platform was built to verify the signal transmission capability of the MMCS system in an air medium.

The results of the test, presented in Figure 11, indicate that the signal attenuated rapidly as the distance increased within 1m. However, beyond a communication distance of 4 m, the signal stabilized with a strength of 2pT, which was greater than the limit of detection as shown in Figure 4d, excluded the interference of a system error, confirming that the received signal was the original signal radiated by the transmitting ME antenna. The MMCS achieved successful signal transmission and reception tests up to a distance of 16 m, and the observed attenuation patterns aligned with the outcomes of similar studies [30,34].

It is necessary to emphasize that although the ME antenna will have a significant signal enhancement at the resonant peak, the resonant frequency of the ME antenna is high, which is not favorable for underwater communication; so, the resonant frequency of the receiver coil was selected as the operating frequency of the MMCS.

### 4.4. Packaging and Underwater Experiment

The packaging structure of the transmitting ME antenna was made using a 3D printing process with a double-deck cylindrical PETG-CF (Polyethylene Terephthalate Glycol-Modified Carbon Fiber) material. The gap between the inner and outer layers of the structure were filled with epoxy resin potting adhesive to ensure waterproofing and corrosion resistance. The final packaged transmitting ME antenna is depicted in Figure 12a. In order to assess the performance of the MMCS in a marine environment, both the transmitting ME antenna and the receiving coil were completely submerged in a simulated seawater solution with 35% salinity at 25 °C, as shown in Figure 12b. When the transmitting ME antenna emitted a 44.75 kHz sine wave signal, the receiving coil, located 1m away from the transmitting ME antenna, detected the electromagnetic wave, and showed the output voltage waveform by oscilloscope, as illustrated in Figure 12c.

When the electromagnetic wave propagated through the simulated saltwater medium, the frequency information experienced minimal shifting. However, compared to the signal received in an air medium, the uniformity of the signal amplitude was poorer. This phenomenon could be attributed to the inadequate uniformity of the simulated seawater solution or the interference of electromagnetic waves from different propagation paths.

## 5. Conclusions

The proposed MMCS operates at a frequency of 44.75 kHz. The system is capable of detecting an electromagnetic field as weak as 1pT and carrying on the binary code element transmission. The MMCS has an input voltage range of 3 V to 200 V, and increasing the input voltage within this range leads to a noticeable improvement in the communication signal strength. By integrating small NdFeB magnets into the transmitting ME antenna, there is a significant enhancement in the communication signal at close distances. In an air medium, the MMCS demonstrates the ability to transmit and receive 44.75 kHz sinusoidal electromagnetic waves over a distance of 16 m. When tested in a simulated seawater medium with 35% salinity at 25 °C, the receiving coil located 1 m away from the transmitting ME antenna successfully receives the electromagnetic wave.

However, the 16 m air communication distance may not be sufficient for certain low-frequency communication applications, and underwater communication distance is expected to reach 30 m. To overcome this limitation, in the future we will focus on more efficient magnetoelectric heterojunction materials, the design of highly sensitive miniaturized receiver devices, and ensure that the transmitting and receiving devices operate at the same resonant frequency. By incorporating these advancements, the communication distance of unit miniaturized communication systems can be improved. At the same time, future work will be devoted to an array of unit miniaturized communication systems to improve the communication distance. Therefore, the MMCS holds significant potential for applications in underground, underwater, and earth–ionosphere waveguide medium communication, as well as energy supply applications.

## Figures and Tables

**Figure 1 micromachines-14-01830-f001:**
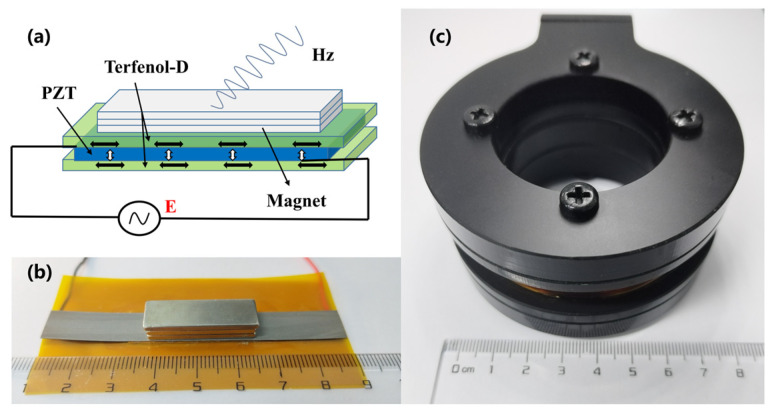
(**a**) Schematic of the MMCS transmitting ME antenna. (**b**) Prototype of the MMCS transmitting ME antenna. (**c**) MMCS receiving coil.

**Figure 2 micromachines-14-01830-f002:**
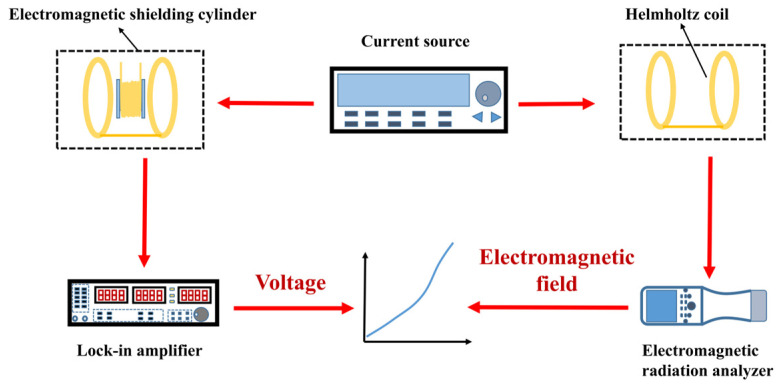
Characterization platform schematic of the receiving coil.

**Figure 3 micromachines-14-01830-f003:**
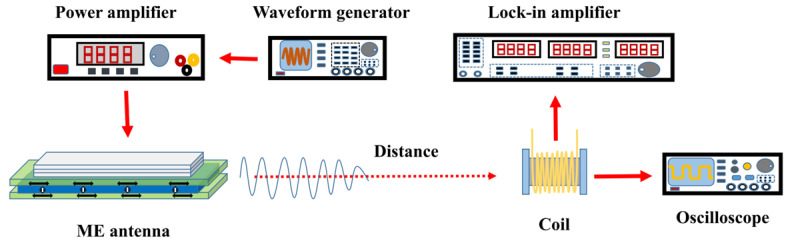
Characterization platform schematic of MMCS.

**Figure 4 micromachines-14-01830-f004:**
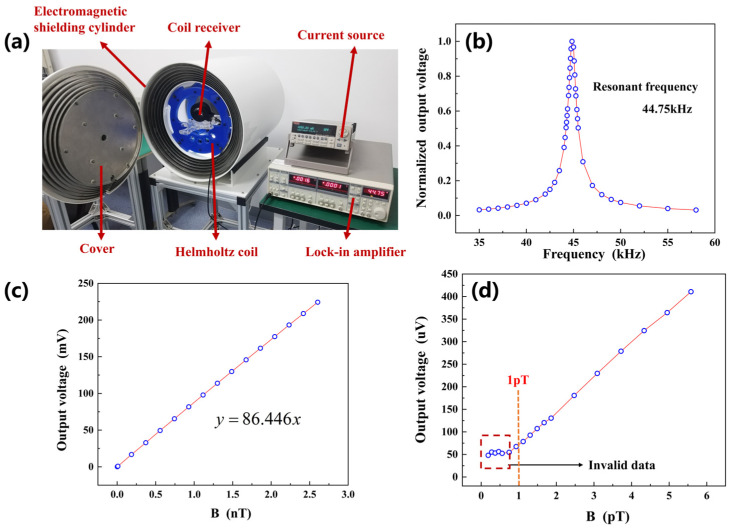
(**a**) Characterization platform of the MMCS receiving coil. (**b**) The receiving coil resonance frequency is 44.75 kHz, and the signal strength at the resonance frequency is improved significantly. (**c**) For the receiving coil, a 1nT electromagnetic field strength corresponds to a 86.446 mV induced voltage at 44.75 kHz. (**d**) The receiving coil resonance frequency detecting an AC electromagnetic field of minimum 1 pT.

**Figure 5 micromachines-14-01830-f005:**
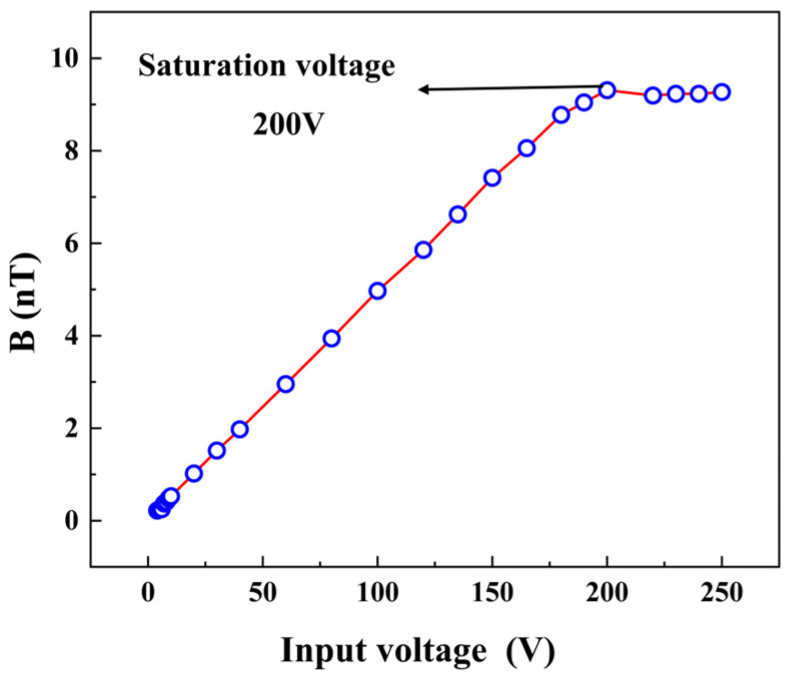
The output signal of the transmitting ME antenna increases with the input voltage, and there are starting and saturation voltages.

**Figure 6 micromachines-14-01830-f006:**
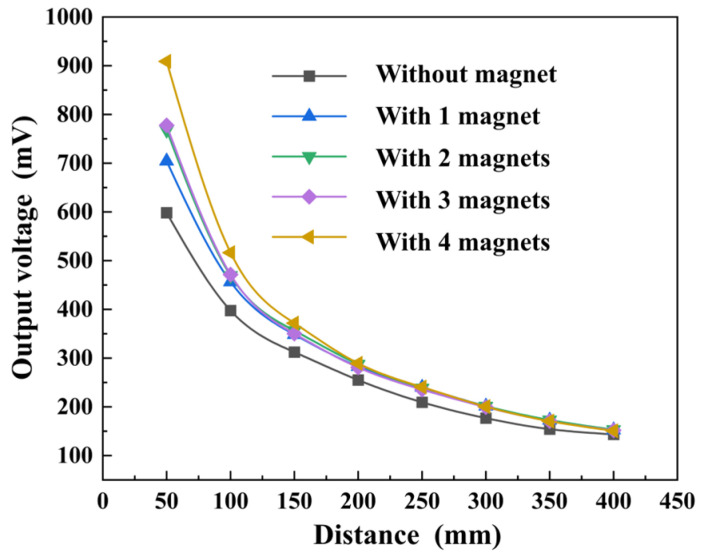
Integration of small NdFeB magnets in the transmitting ME antenna helps to improve the radiated signal strength.

**Figure 7 micromachines-14-01830-f007:**
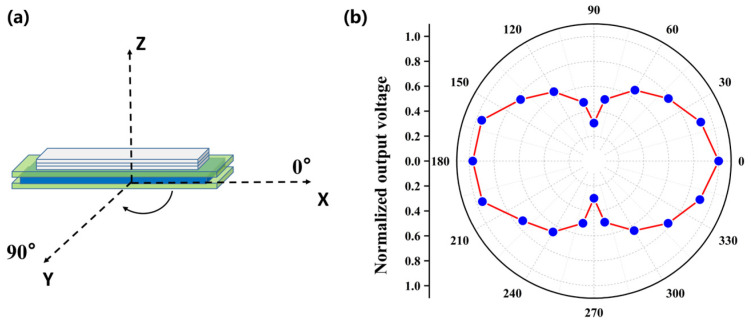
(**a**) Coordinate system definition diagram of the ME antenna. (**b**) Radiating pole diagram of the ME antenna in the *x*-*y* plane.

**Figure 8 micromachines-14-01830-f008:**
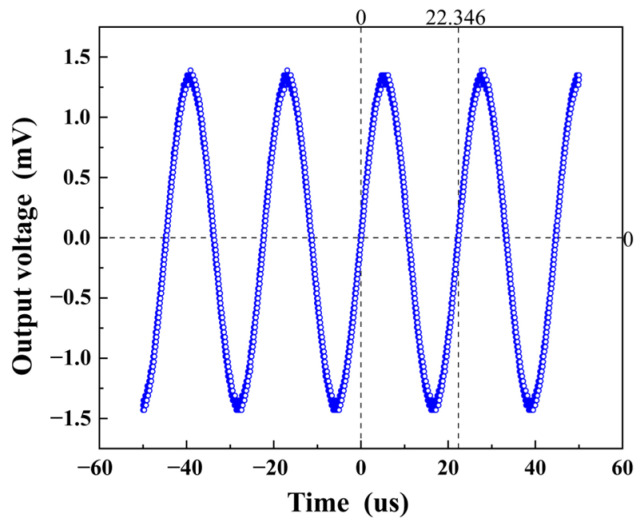
Receiving coil output 44.75 kHz sine wave voltage consistent with the radiated signal of the transmitting ME antenna.

**Figure 9 micromachines-14-01830-f009:**
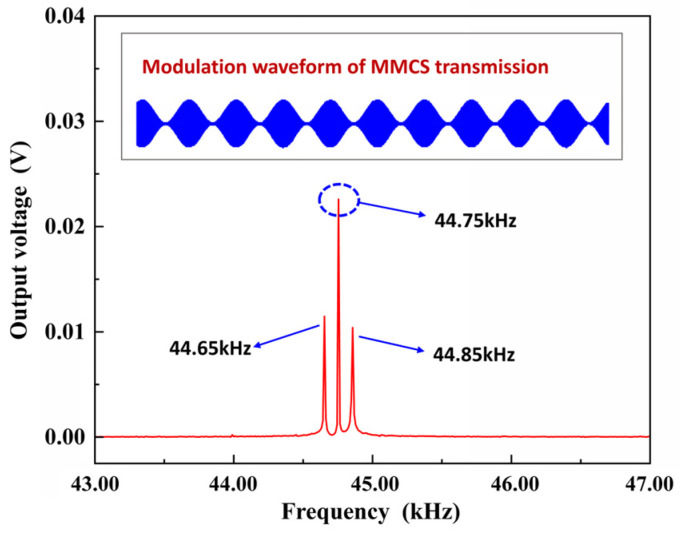
Binary code element transmission experiments of the MMCS.

**Figure 10 micromachines-14-01830-f010:**
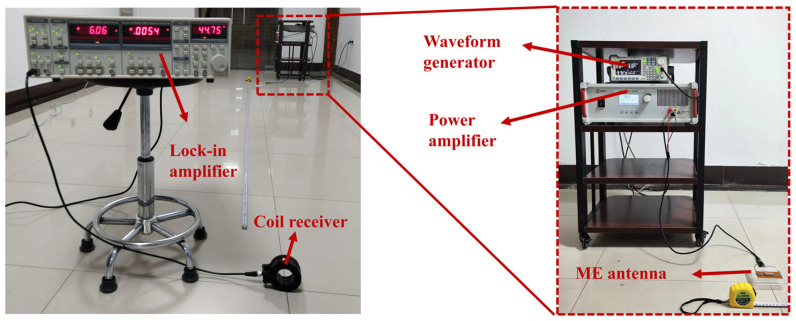
The testing platform of the MMCS in air.

**Figure 11 micromachines-14-01830-f011:**
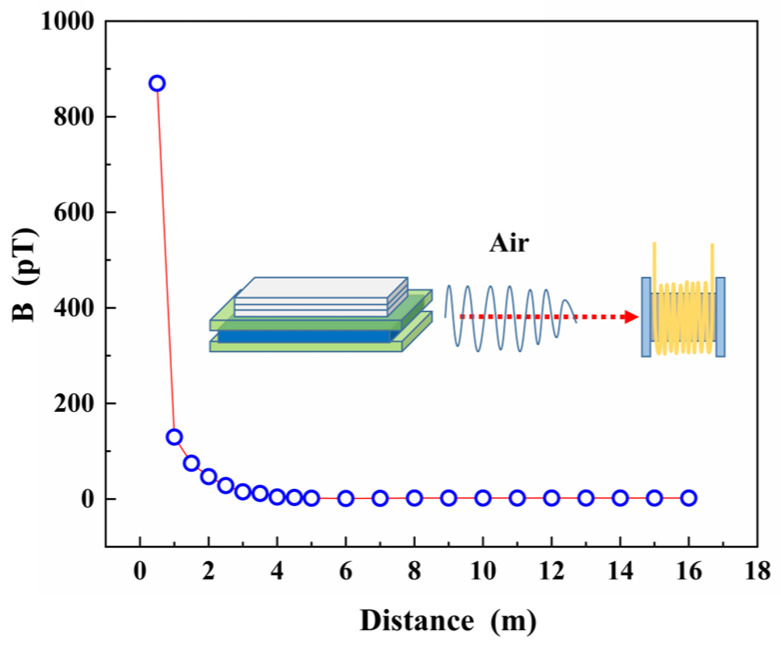
MMCS signal strength as a function of distance in the actual test process can achieve communication up to 16 m.

**Figure 12 micromachines-14-01830-f012:**
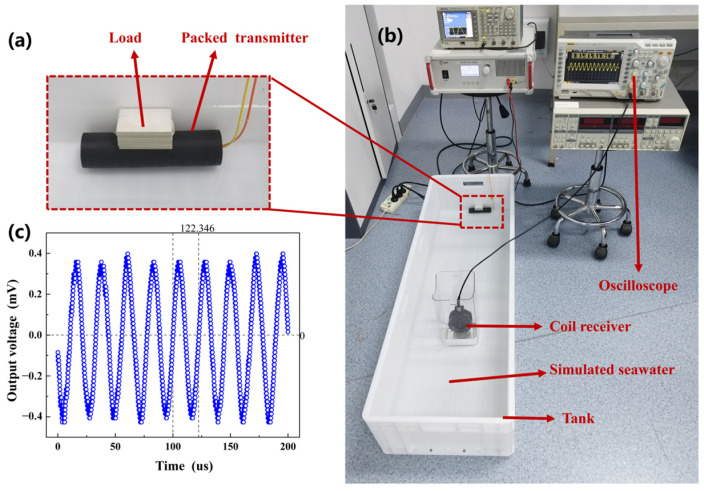
Simulation of seawater low-frequency communication experiment (**a**) Packaged waterproof transmitting ME antenna. (**b**) Simulation of seawater medium communication feasibility test. (**c**) Simulation of received signal amplitude in seawater medium.

## Data Availability

The data presented in this study are available on request from the corresponding author. The data are not publicly available due to privacy-related legal issues.

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
