# Peer review of "Miniaturized Low-Frequency Communication System Based on the Magnetoelectric Effect"

_micromachines, 2023, doi:10.3390/mi14101830_

Round 1
Reviewer 1 Report (New Reviewer)
In this paper, a miniaturized magnetoelectric communication system is introduced. The performances of the proposed system are characterized via a ground test and an underwater test. The content of this manuscript is good.
1. On Page 3, the authors mentioned that the proposed MMCS showed great potential in underwater applications. But the communication range is only 1 meter during the test. I think it is not enough to show the potential since the range is too small.
2. On Page 5, the authors mentioned that the operating bandwidth of the receiving coil is 1.1kHz. How did the authors determine the bandwidth?
3. In Fig.6, it can be observed that when the distance increases, the output voltage of ME antennas with different NdFeB magnets merges. The authors need to make some explanation here.
4. On Page 7, the authors mention that the signal stabilizes with a strength of 2pT, which is greater than the limit of detection. What is the limitation of the detection? Or what is the minimum value the system can handle?
5. On Page 7, the authors mention the enhancement will become im-perceptible with the increase of distance. What does it mean? Do the authors want to claim that the enhancement is useless? Does it make sense?
6. On Page 7, the authors mention since the ME antenna works at non-resonant frequency, it significantly expands the transmitting bandwidth to meet the transmitting demand of different frequency bands. I think more explanations are needed here to make it more clear. Why the bandwidth is expanded.
Author Response
Please see the attachment.

Reviewer 2 Report (New Reviewer)
This paper presents the design of a low frequency miniaturized magnetoelectric communication system including transmitter device and receiver device. This paper was well written. Experimental design, result, discussion are complete, reasonable, and clear. This topic is interesting for readers. The testing platform of MMCS is good design in air. MMCS signal strength as a function of distance, in the actual test process can achieve 16m communication. However, when tested in a simulated seawater medium with 35% salinity at 25℃, the receiving coil located 1m away from the transmitting ME antenna successfully receives the electromagnetic wave. I think that these are good findings and results. Therefore, I recommend this article can be accepted for publication in Micromachines without revision.
Author Response
Thank you very much for your recognition so that our article can be seen by more readers. In future articles, we will work on improving the performance of the equipment.
Reviewer 3 Report (New Reviewer)
Comments:
In the paper titled as “Miniaturized low-frequency communication system based on magnetoelectric effect”, the authors proposed a low-frequency communication system based on magnetoelectric devices, and characterized the device performance. Before its publication, there are many issues should be raised.
1. The paper only shows electromagnetic wave transmission tests and does not see tests of communication performance (binary code element transmission), which cannot be described as a communication system.
2. The paper does not compare the performance with the low-frequency mechanical antennas reported in recent years.
3. The device structure in this paper is very similar to the ME device design scheme of references 34 and 35. However the operating frequencies of the transmitter and receiver are not matched.
4. In the statement on the second page regarding reference 29, the name of the person should be written as Dong.
5. In the first paragraph on page 3, you suggest that integrating permanent magnets improves the efficiency of stress transfer. May I ask how to improve it?
6. In the second paragraph on page 3, you mention that the device is operating in L-T mode. So why is the device oriented toward the coil in the width direction instead of the length direction in Figure 8?
7. In the third paragraph on page 4, you mention an electromagnetic radiation analyzer, how does its detection limit compare to that of a resonant receiver coil? Please give a conversion curve of the Helmholtz coil drive current to the target magnetic field and add a description of the key metrics of this product.
8. In Fig. 3, the lengthwise direction of the device is toward the coil, but in Fig. 8 it is the widthwise direction of the device that is toward the coil? Why is the coil perpendicular to the direction of radiation instead of parallel to it?
9. In the first paragraph on page 5, how is the bandwidth 1.1 kHz defined?
10. In the second paragraph on page 5. There is a problem with the expression, it should be the transfer function at that operating frequency. Why don't you sweep the resonant frequency of the device?
11. In the last paragraph on page 5, has any consideration been given to the power capacity limitations of ceramics? Have any calculations been made for their maximum carrying voltage?
12. On page 6, first paragraph, you say " This enhancement is attributed to the pressure effect of the magnets on the bonding layer of the transmitting ME antenna, rather than the DC bias magnetic field. Terfenol-D is effectively polarized and is not responsive to bias magnetic fields.". But you actually changed the size of the bias field while changing the number of permanent magnets. It is incorrect to say that the device is not affected by the bias magnetic field.
13. In the first paragraph on page 7, there is a problem with the presentation here.
14. In paragraph 3 on page 7, the fact that the BNC cable was not used on the transmitter side of the experiment does not rule out that it was interference from the system feeder and a blank control experiment should have been done.
15. Please change Figure 9 to a logarithmic coordinate system.
16. Please provide the radiating pole diagram of the antenna in Figure 10.
17. Please double-check the citation information of the references. Reference 40 should be published in Physics Letters A.
In summary, this paper still has many technical problems, major revisions must be made.
-
English writing should be polished.
Round 2
Reviewer 1 Report (New Reviewer)
All the comments have been addressed.
Author Response
Thank you for your recognition.
Reviewer 3 Report (New Reviewer)
Comments:
The authors have made some revisions, but there are some minor corrections should be made. Following as:
1. Does 200V in Figure 5 refer to the effective value or peak to peak value?
2. Please change both Figure 6 and Figure 11 to double logarithmic coordinate axis drawing.
3. Which plane is the radiation pole diagram in Figure 7 located in?
No comments.
Author Response
Please see the attachment.

This manuscript is a resubmission of an earlier submission. The following is a list of the peer review reports and author responses from that submission.
Round 1
Reviewer 1 Report
This reviewer greets the authors for the present work that is clearly presented and well written. Some remarks about the manuscript:
This reviewer notices a lack of comparison of the present work with similar systems present in the literature. A more prominent comparison would highlight its qualities.
Does the fabricated magnetoelectric antenna operates at its mechanical resonant frequency? It is not clear from the text and what advantages or disadvantages this would bring.
Please define the acronym “L-T” in “ L-T vibration mode”, “PI” in “a pair of PI substrate electrodes” and “PZT” in “polarized PZT piezoelectric sheet” in the text.
“The transmitting ME antenna requires a starting voltage, and when the excitation voltage is below 3V, the vibration of the PZT sheet is weak,” The excitation voltage that is defined to be between 3V and 200V, is it a DC polarizing voltage that is summed with the sine wave signal or is it related solely to the amplitude of the sine wave signal? It is not clear in the text.
It is unknown to this reviewer that some implementations of Bluetooth exists in the millimeter wave spectrum (30 GHz – 300 GHz). Could the authors provide specific references in the text that are clearer on this matter?
Reviewer 2 Report
The work is an experimental study. Novelty is perhaps only at simulation and experimental layout. This must be strengthen.
Its an interesting work, with potential practical applications. But in its present, seems not usable. It is an effort at beginning, this is not necessary bad, but, seems that there are many things that have to be clarified. For example. Is this device going to be used for underwater communication between divers? Seems to be so, among others. Then a safety analysis is needed, especially if rather high voltages are going to be used.
What the target distances inside the water, or practical distances, that must be achieved, in order to have a devise which can be used? Is that a near future target?
Only the transmitting antenna and the receiving coil were submerged in the water. What about the other components in their final shape? How big are going to be? At title and at main body the device is presented as Miniaturized magnetoelectric communication system. In my opinion only the antennas are presented. All the other components are laboratory instruments.
Are the authors ready to provide an ready to be used device, even with limitations?
Reviewer 3 Report
Authors in this research work have presented a MMCS operating at a frequency of 44.75 kHz. It consists of a transmitting ME antenna with dimensions of 80mm×10mm×3mm and a receiving device in the form of a manually crafted coil. The system is capable of detecting an electromagnetic field as weak as 1pT. From this reviewer’s point of view, the topic and content of this paper were found interesting. The promising results have been achieved and evaluated in a well-organized manuscript, also an experimental validation was provided. Although this paper seems attractive for readers in the field of micromachines, authors are requested to address the following comments to improve its quality prior to final recommendation.
1) What do you mean about the communication systems in the title? Please add the name of the device, such as “transmitter device”.
2) The first three sentences of the abstract section are unnecessary and they can be removed. Therefore, in this part please try to add more information about the design process of the proposed transmitter device, its antenna, and its other sections?
3) More numerical results can be added to the abstract section.
4) There are various methods such as metamaterial and metasurfaces to improve the performance parameters of the communication systems specially antenna devices and design them in a compact footprint area. Therefore, these methods can be briefly mentioned in the introduction section to improve this part. Below are helpful suggestions.
“New compact printed leaky‐wave antenna with beam steering”, Microwave and Optical Technology Letters 58 (1), 215-217, 2016.
“Improved adaptive impedance matching for RF front-end systems of wireless transceivers”, Scientific Reports 10 (14065), 1-11, 2020.
“Super-wide impedance bandwidth planar antenna for microwave and millimeter-wave applications”, Sensors 19 (10), 2306, 2019.
“Design and Realization of a Frequency Reconfigurable Antenna with Wide, Dual, and Single-Band Operations for Compact Sized Wireless Applications”, Electronics 10 (11), 1321, 2021.
“Dual-Polarized Highly Folded Bowtie Antenna with Slotted Self-Grounded Structure for Sub-6 GHz 5G Applications”, IEEE Transactions on Antennas and Propagation 70 (4), 3028-3033, 2022.
“A new miniature ultra wide band planar microstrip antenna based on the metamaterial transmission line”, 2012 IEEE Asia-Pacific Conference on Applied Electromagnetics (APACE), 293-297, 2012.
5) MMCS transmitting ME antenna has been shown in Fig.1, please explain its design process in depth. Its feeding principle also needs to be elaborated.
6) The quality of the plots needs to be improved. The font size of the texts inside the figures are small and it is difficult to read them.
7) How have authors achieved the received signal amplitude in seawater medium which is plotted in Fig.10(c)? Please explain it.
8) Conclusion is too long, please summarize it.
9) Reference part needs to be improved by a proper extension as per above mentioned suggestions.
10) Language of the paper can be slightly polished.
Language of the paper can be slightly polished.
Reviewer 4 Report
Authors propose a miniaturized low-frequency communication system operating at a frequency of 44.75 kHz
There are some issues that must be fixed before finally accepting the paper:
- Authors must explain the structure of the paper at the end of the introduction section.
- Authors should test if the communication system depends on the conductivity/salinity and the temperature of the water.
- Author should test the bandwidth of the proposed communication system.
- Authors should include their future work at the end of the conclusion section.
Round 2
Reviewer 2 Report
The work is interesting but not mature yet.